# Veno-Arterial Extracorporeal Membrane Oxygenation (ECMO) Impairs Bradykinin-Induced Relaxation in Neonatal Porcine Coronary Arteries

**DOI:** 10.3390/biomedicines10092083

**Published:** 2022-08-25

**Authors:** Livia Provitera, Giacomo S. Amelio, Matteo Tripodi, Genny Raffaeli, Francesco Macchini, Ilaria Amodeo, Silvia Gulden, Valeria Cortesi, Francesca Manzoni, Gaia Cervellini, Andrea Tomaselli, Gabriele Zuanetti, Caterina Lonati, Michele Battistin, Shady Kamel, Valeria Parente, Valentina Pravatà, Stefania Villa, Eduardo Villamor, Fabio Mosca, Giacomo Cavallaro

**Affiliations:** 1Neonatal Intensive Care Unit, Fondazione IRCCS Ca’ Granda Ospedale Maggiore Policlinico, 20122 Milan, Italy; 2Department of Clinical Sciences and Community Health, Università degli Studi di Milano, 20122 Milan, Italy; 3Department of Pediatric Surgery, Fondazione IRCCS Ca’ Granda Ospedale Maggiore Policlinico, 20122 Milan, Italy; 4Department of Pediatric Surgery, ASST Grande Ospedale Metropolitano (GOM) Niguarda, 20162 Milan, Italy; 5Department of Pathophysiology and Transplantation, University of Milan, 20122 Milan, Italy; 6Center for Preclinical Investigation, Fondazione IRCCS Ca’ Granda Ospedale Maggiore Policlinico, 20122 Milan, Italy; 7Betamed Perfusion Service, 00192 Rome, Italy; 8Transfusion Center and Blood Component Bank of Rare Groups, Fondazione IRCCS Ca’ Granda Ospedale Maggiore Policlinico, 20122 Milan, Italy; 9Department of Pediatrics, Maastricht University Medical Center (MUMC+), School for Oncology and Reproduction (GROW), University of Maastricht, 6229 Maastricht, The Netherlands

**Keywords:** extracorporeal membrane oxygenation (ECMO), endothelial dysfunction, porcine coronary arteries, bradykinin

## Abstract

Extracorporeal membrane oxygenation (ECMO) is a lifesaving support for respiratory and cardiovascular failure. However, ECMO induces a systemic inflammatory response syndrome that can lead to various complications, including endothelial dysfunction in the cerebral circulation. We aimed to investigate whether ECMO-associated endothelial dysfunction also affected coronary circulation. Ten-day-old piglets were randomized to undergo either 8 h of veno-arterial ECMO (*n* = 5) or no treatment (Control, *n* = 5). Hearts were harvested and coronary arteries were dissected and mounted as 3 mm rings in organ baths for isometric force measurement. Following precontraction with the thromboxane prostanoid (TP) receptor agonist U46619, concentration–response curves to the endothelium-dependent vasodilator bradykinin (BK) and the nitric oxide (NO) donor (endothelium-independent vasodilator) sodium nitroprusside (SNP) were performed. Relaxation to BK was studied in the absence or presence of the NO synthase inhibitor Nω-nitro-L-arginine methyl ester HCl (L-NAME). U46619-induced contraction and SNP-induced relaxation were similar in control and ECMO coronary arteries. However, BK-induced relaxation was significantly impaired in the ECMO group (30.4 ± 2.2% vs. 59.2 ± 2.1%; *p* < 0.0001). When L-NAME was present, no differences in BK-mediated relaxation were observed between the control and ECMO groups. Taken together, our data suggest that ECMO exposure impairs endothelium-derived NO-mediated coronary relaxation. However, there is a NO-independent component in BK-induced relaxation that remains unaffected by ECMO. In addition, the smooth muscle cell response to exogenous NO is not altered by ECMO exposure.

## 1. Introduction

Extracorporeal membrane oxygenation (ECMO) is a lifesaving procedure for children and adults with reversible respiratory and/or cardiovascular failure refractory to standard treatment [1,2]. ECMO remains essential in several conditions as a bridge to recovery, transplantation, or other therapeutical decisions [3,4]. Indications and criteria for ECMO have changed over the years, but it continues to be the only option for some patients failing other medical therapies [5,6]. Since 1990, more than 45,000 newborns have been supported with ECMO. Patient outcomes have improved over time, with survival to discharge or transfer ranging from 73% and 43% to 42% for respiratory, cardiac, or cardiopulmonary resuscitation indications, respectively [7,8]. Although ECMO technology and expertise have increased throughout the years, the procedure is still burdened by complications, affecting short- and long-term outcomes, including bleeding and neurological complications [7,9].

ECMO initiation is associated with an immediate and complex inflammatory reaction, similar to that seen in systemic inflammatory response syndrome (SIRS) [10]. This SIRS-like reaction induces oxidative stress and plays a central role in the pathogenesis of most ECMO complications [10,11,12]. The exposure of patients’ blood to the extracorporeal surface leads to the activation of coagulation and inflammatory cascades [10]. Typically, 1–2 h after beginning the procedure, the contact between blood and components in the ECMO circuit triggers complement-activating factors, which leads to the activation of neutrophils and release of cytokines [13]. In addition, neutrophils produce different cytolytic proteins, such as elastase, myeloperoxidase, and lysozyme [14,15]. The massive activation of the pro-inflammatory and oxidative systems is very damaging, especially if the antioxidant defenses are incompletely developed, as is frequency the case in the neonatal period [16]. In addition, there is a growing recognition that ECMO exposure induces vascular endothelial dysfunction [17,18]. However, studies on the effects of ECMO on endothelial function have so far been limited to cerebral circulation [17,18].

Our hypothesis in the present study is that ECMO-induced endothelial dysfunction is not limited to cerebral vessels. To test this hypothesis, we conducted an ex vivo study of endothelial function in coronary arteries from 10-day-old piglets exposed to 8 h of Veno Arterial (VA) ECMO. Endothelium-dependent relaxation was assessed by the response to bradykinin (BK), a widely used mediator in evaluating endothelial function in coronary arteries [19,20,21,22,23,24]. Moreover, BK levels increase markedly during ECMO and it has been suggested that BK plays a key role in the pathophysiology of ECMO-associated complications [10,25].

## 2. Materials and Methods

The experiments were conducted on 10-day-old Large White pigs (*n* = 10), weighing between 4 and 6 kg and randomized into two groups: ECMO (*n* = 5) and control (*n* = 5).

All animal procedures were carried out in accordance with the national and European legislative and administrative provisions in force. The study protocol was approved by the animal welfare body (OPBA) of the Università degli Studi di Milano and by the Italian Ministry of Health (authorization number 15/2015 PR).

### 2.1. Blood Collection for Extracorporeal Priming Circuit

Blood for extracorporeal priming circuit or transfusions was collected 24 h before at the slaughterhouse and stored at 4 °C in sterilized 500 mL glass bottles. Citric dextrose acid (70 mL) was added to each bottle as anticoagulant. In addition, vancomycin (25 mg) and gentamicin (20 mg) were added to 500 mL of blood to reduce the risk of infection. Before proceeding with the priming of the circuit, blood compatibility tests were performed with the animal under study. Direct blood group typing was performed by manual method, using ABO Ortho BioVue Card (Ortho Clinical Diagnostics, Pencoed, UK) on 3% blood cells diluted in saline solution. Indirect blood group typing was manually performed on fresh plasma through Reverse Diluent Ortho BioVue Card with standard blood cells A1, B, and 0 (Affirmagen, Ortho Clinical Diagnostics, Pencoed, UK). We repeated the same analyses at 4 °C to enhance ABO antibody-mediated activity. Major compatibility (recipient serum versus donor red blood cells) and minor compatibility (donor plasma with the recipient red cells) were manually performed using Ortho BioVue with anti-IgG, C3d polyspecific Cards.

The donor animal was chosen based on the negativity of both compatibility tests.

### 2.2. Surgical Procedures

Animals in both groups were sedated by an intramuscular (i.m.) administration of medetomidine (0.025 mg/kg) and tiletamine/zolazepam (5 mg/kg). In addition, the lateral auricular vein was cannulated by a 24 G cannula in all animals.

After ear vein cannulation, animals belonging to the control group were euthanized. Animals in the ECMO group were anesthetized with a continuous intravenous (i.v.) infusion of propofol (5 mg/kg/h), medetomidine (10 μg/kg/h), and rocuronium (7 μg/kg/min), surgically tracheostomized, intubated, and ventilated (Baby Log 8000 Plus; Draeger; Germany). The left carotid artery and left external jugular vein were cannulated for blood pressure monitoring, blood sampling, and infusion of medications. A surgical cystostomy with a 6 French Foley catheter (Coloplast A/S, 3050 Humlebaek, Denmark) was made to monitor the diuresis during the extracorporeal procedure. In addition, i.v. ceftriaxone (2 g) was administered as antibiotic therapy to prevent the onset of infectious complications.

Cannulation of the right carotid artery with an 8 Fr catheter (Medtronic, Minneapolis, MN, USA) was achieved by placing the catheter up to the aortic arch, while cannulation of the external jugular vein with a 10 Fr catheter (Medtronic, Minneapolis, MN, USA) was established and advanced into the right atrium by pediatric surgeons.

All animals were systematically heparinized with a loading dose of heparin (100 IU/kg) following a heparin maintenance dose (25 IU/kg/h).

### 2.3. ECMO Procedure

A polyvinyl chloride 1/4 (external diameter) × 1/16 (tube thickness) inch tubing circuit and RotaFlow centrifugal blood pump with tip-to-tip BIOLINE coating and polymethylpentene oxygenator (Permanent Life Support (PLS) System and Quadrox iD lung—Pediatric (Getinge Group, Gothenburg, Sweden)) was used for VA ECMO. Body temperature was maintained at 38 °C during the extracorporeal procedure with a heater unit HU 35 (Getinge Group, Gothenburg, Sweden). The CardioHelp System (Getinge Group, Gothenburg, Sweden) monitored continuous venous and arterial pressure control. Temperature (°C), hemoglobin (Hb, g/dL), hematocrit (Hct, %), venous and arterial oxygen saturation (SvO_2_, SaO_2_, %), oxygen delivery (DO_2_, mL/min), oxygen consumption (VO_2_, mL/min), oxygen extraction ratio (O_2_ER), oxygen partial pressure (pO_2_, mmHg), and carbon dioxide partial pressure (pCO_2_, mmHg) were continuously monitored by Spectrum M4 (SpectrumMedical, Gloucester, UK) (Figure 1) [26,27].

As previously described, the extracorporeal circuit was primed with 125 mL of porcine blood. The extracorporeal blood flow was fixed at 80–100 mL/kg/min with 0.25 L/m of sweep gas to obtain an arterial pCO_2_ (PaCO_2_) 50 mmHg. The activated coagulation time (ACT) was maintained between 180 and 200 s. The ventilator was set to deliver a tidal volume of 7 mL/kg, PEEP 5 cm H_2_O, FiO_2_ 0.3, respiratory rate 30 breaths/minute. All these parameters were kept constant during the experiment. An electrolyte solution (100 mL/kg) and a 10% glucose solution (20 mL/kg) were infused during the entire experiment.

Vital parameters were continuously recorded while blood gas analysis and ACT were performed every 15 min in the first hour of ECMO, every 30 min in the second hour, and every hour until the end of the experiment, 8 h from the beginning of the support (Figure 2).

### 2.4. Euthanasia and Tissue Collection

The animals in both groups were euthanized by medetomidine (50 μg/kg) and propofol (10 mg/kg) overdose, followed by intravenous potassium chloride (20 mEq). The heart was quickly removed after sacrifice and placed in the cold Krebs-Ringer bicarbonate (KRB) solution.

### 2.5. Vascular Reactivity Studies

The left anterior descending coronary artery was carefully dissected free of surrounding tissue and cut into 2–3 mm length rings under a dissection microscope. After dissection, two L-shaped stainless-steel wires were inserted into the arterial lumen and the rings were suspended in 5 mL organ baths filled with KRB solution maintained at 37 °C and aerated with 95% O_2_-5% CO_2_ (pH 7.4), as previously described [28,29,30]. One wire was attached to the chamber and the other to an isometric force-displacement transducer (model PRE 206-4, Cibertec, Madrid, Spain). The isometric force signal was amplified, A/D converted and recorded (MP100 data acquisition system, BIOPAC System Inc. Santa Barbara, CA, USA). As determined from previous experiments, an optimal resting tension of 1 g (9.8 mN) was applied to the rings [30]. Tissues were allowed to equilibrate for 60–90 min. During this period, they were restretched and washed every 30 min with warm KRB solution. Before starting the experiments, the rings were transiently challenged with 62.5 mM KCl for 10 min to assess the functional state and establish a reference nonreceptor-mediated contraction to standardize contractile responses. Next, the rings were washed three times and then allowed to rest for 30 min.

Relaxant responses were studied following 10 min of contraction of the vessels with the thromboxane prostanoid (TP) receptor 9,11-Dideoxy-9a,11a-methanoepoxy prostaglandin F_2a_ (U46619, 1 μM). This concentration elicited ~80% of the maximal contractile response to the drug in the coronary artery, as determined in previous experiments [30]. When contraction induced by U46619 reached a plateau, concentration–response curves were performed for the different relaxant agents. The agent concentration was incremented once the response had reached a plateau or after 5–10 min if no response had occurred or a clear plateau was not reached.

The following relaxant agents were tested: the non-selective bradykinin receptors (B_1_ and B_2_) agonist BK (10 pM—1 μM) as endothelium-dependent vasorelaxation and the NO donor sodium nitroprusside (SNP, 10 pM—1 μM) as endothelium-independent vasorelaxation. At the end of the concentration–response curve, we administered papaverine hydrochloride (0.1 mM) for 10 min to assess whether the rings had already reached maximum relaxation.

Some experiments were conducted with the inclusion of NO synthase (NOS) inhibitor Nω-nitro-L-arginine methyl ester hydrochloride (L-NAME, 0.1 mM) for 30 min.

### 2.6. Drugs and Solutions

KRB buffer is composed as follows (in mmol L^−1^): NaCl, 118.5; KCl, 4.75; MgSO_4_·7H_2_O, 1.2; KH_2_PO_4_, 1.2; NaHCO_3_, 25.0; CaCl_2_, 2.5; glucose, 11.1. Solutions including different concentrations of KCl were prepared by adding KCl instead of NaCl in an equimolar amount.

U46619, SNP, and papaverine were obtained from Sigma-Aldrich Chemical Co (St. Louis, MO, USA). All the other drugs were obtained from Tocris (Ballwin, MO, USA).

Papaverine hydrochloride, BK, L-NAME, and SNP were dissolved in distilled deionized water. U46619 was dissolved in methyl acetate. The vehicles’ concentration did not exceed 0.1% and did not influence the mechanical activity of the vessels.

### 2.7. Data Analysis

Results are shown as mean ± standard errors of means (S.E.M.). Contraction responses with U46619 were expressed as a percentage of contractions induced by KCl.

The relaxant responses with BK and SNP were expressed as a percentage of papaverine-induced relaxation.

Sensitivity/potency (expressed as pD2 = −log EC_50_) and maximal relaxation (Emax) to agonists were determined by fitting individual concentration–response data to a nonlinear sigmoidal regression curve.

Mean value differences were calculated by Student’s *t*-test or two-way ANOVA followed by Bonferroni’s post hoc *t*-test.

Differences were considered significant at a *p* < 0.05. GraphPad Prism was the software used for all the analyses performed (version 9.0.1.151 for Windows, GraphPad Software, San Diego CA, USA).

## 3. Results

### 3.1. ECMO Procedure

Ten animals were equally randomized into two groups. Age (Control group: 9.6 ± 1.28 days; ECMO group: 9.2 ± 1.16 days), sex (Control group: female 5; ECMO group: female 5), and weight (Control group: 6.44 ± 0.16 kg; ECMO group: 6.28 ± 0.12 kg) in each group were similar. No statistically significant differences were found between the two groups.

In ECMO animals, physiological variables were maintained during the study period. The mean body temperature was 37.1 ± 1.1 °C, mean arterial oxygen saturation was 97.7 ± 1.7%, and mean systolic arterial pressure was 80–100 ± 10 mmHg (Table 1).

### 3.2. Reactivity of Coronary Arteries

KCl and U46619 induced a tonic contraction of the arteries that was not significantly different between the ECMO and the control group (Figure 3).

BK and SNP relaxed U46619-contracted vessels in a concentration-dependent manner. ECMO-treated vessels showed a significantly reduced maximal response (E_max_) to BK as compared to control vessels (*n* = 5; *p* < 0.0001; two-way ANOVA) (Figure 4, Table 2). However, the sensitivity (pD2) to BK was not significantly different between ECMO-exposed and control vessels. SNP-induced relaxation was similar in ECMO-exposed and control coronary arteries (Figure 5, Table 2). In order to assess whether the reduced response to BK in ECMO coronary artery rings could be due to the reduced bioavailability of endothelium-derived NO, additional experiments in the presence of the NOS inhibitor L-NAME were performed. After 30 min of incubation with L-NAME (0.1 mM), we observed no difference between the control and ECMO groups on the relaxation induced by BK (Figure 6, Table 2).

## 4. Discussion

To the best of our knowledge, the present study is the first to investigate the effects of ECMO exposure on vascular reactivity in vessels not supplying the central nervous system. Coronary arteries of 10-day-old piglets exposed to 8 h of veno-arterial ECMO showed a significant impairment in BK-mediated relaxation. In contrast, the relaxation mediated by the NO donor SNP was not affected by ECMO exposure. Interestingly, the presence of the NOS inhibitor L-NAME impaired BK-mediated relaxation in the control group to a level comparable to the relaxation observed in ECMO-exposed coronary arteries. Taken together, our data suggest that ECMO exposure impairs endothelium-derived NO (EDNO)-mediated coronary relaxation. However, there is an EDNO-independent component in BK-induced relaxation that remains unaffected by ECMO (Figure 7). Further, the smooth muscle cell response to exogenous NO is not altered by ECMO exposure.

A large body of literature has explored the physiology and pharmacology of BK and defined two receptor types, B_1_ and B_2_, which mediate its multiple effects [31,32,33]. B_2_ receptors are constitutively expressed, mediating most of the vascular and metabolic actions of BK [31,32,33]. Conversely, B_1_ receptors are generally absent in physiological conditions but may induce various pathological conditions, including inflammation, tissue trauma, and disruption of B_2_ receptors [31,32,33]. The mechanisms involved in BK-induced vascular relaxation vary according to species, vascular bed, and vessel size, but the porcine coronary artery is one of the vessels that has been most extensively studied [19,20,21,22,23,24]. BK induces porcine coronary artery relaxation via endothelial B_2_ receptors. This effect can be blocked partly by NOS inhibitors, such as L-NAME, suggesting a role for de novo synthesis of NO from L-arginine by eNOS [19]. The relaxant effect of BK that is not blocked by NOS inhibitors is attributed to EDHF [20,21,22,23].

Our experimental data suggest that ECMO exposure impairs endothelium-derived NO (EDNO)-mediated coronary relaxation. However, there is an EDNO-independent component of BK-induced relaxation that is attributed to endothelium-derived hyperpolarization factor (EDHF) and remains unaffected by ECMO (Figure 7).

The release of EDHF from endothelial cells depends on the activation of endothelial intermediate-conductance and small-conductance Ca^2+^-dependent K^+^-channels (IK_Ca_, SK_Ca_). Several EDHF candidates have been proposed, but in porcine coronary arteries, *S*-nitrosothiols have the strongest experimental support [20,21,22,23]. Subsequently, EDHF induces smooth muscle cell hyperpolarization through activation of inwardly rectifying K^+^ channel (K_IR_) channels, Na^+^/K^+^-ATPase and/or large-conductance Ca^2+^-dependent K^+^ (BK_Ca_) channels [20,21,22,23]. In addition, NO itself is capable of inducing hyperpolarization via activation of Ca^2+^-dependent K^+^ (K_V_) channels and Na^+^-K^+^-ATPase. Furthermore, BK also releases NO from stores of NO-containing factors, such as *S*-nitrosothiols [20,21,22,23]. Importantly, by reacting with other thiols at physiological pH, S-nitrosothiols yield nitroxyl (HNO), a recently discovered EDHF. HNO acts as a relaxant, either directly or via its conversion to NO [23] (Figure 7).

The main limitations of our study are that it is exclusively based on functional experiments and we did not explore the EDHF pathway. However, as mentioned above, there is substantial experimental evidence for an important role of the EDHF pathway in BK-mediated relaxation in porcine coronary arteries [20,21,22,23]. Moreover, our results are in agreement with those of Thollon et al., who demonstrated that, in porcine coronary arteries subjected to angioplasty, EDNO-mediated relaxation induced by BK is reduced while the EDHF component is not affected [24]. Thus, the production of EDHF explains the maintained relaxation by BK despite the reduced endothelial production of NO [24]. We speculate that the same phenomenon occurs in coronary arteries exposed to ECMO. Nevertheless, the relevance of the EDHF pathway increases as coronary artery size decreases [22,23]. Therefore, ECMO-induced endothelial dysfunction may have a lower impact on coronary microcirculation.

Extracorporeal circulation raises endogenous BK levels to 20-times the basal levels [25]. The ECMO circuit appears to be a decisive factor in this process, as BK levels in the arterial blood leaving the pump were higher than those in the mixed venous blood arriving at the pump [25]. The mechanism is mediated by activation of the contact coagulation pathway with an increase in FXIIa and reduction in C1 inhibitor [10]. The relevance of this process is demonstrated by the anti-inflammatory efficacy of trials with inhibitors of FXIIa and C1 esterase in patients in extracorporeal circulation [34,35]. In addition to plasma kinin increase, the endothelial B1 and B2 receptors are upregulated during inflammation [36]. All these data would suggest an increase in BK-induced effects in the ECMO group. However, our study shows that ECMO-induced vascular impairment leads to a counter-regulatory mechanism, attenuating the relaxant effects of BK.

As mentioned in the Introduction, the effects of ECMO on vascular reactivity have been poorly investigated, except for the cerebrovascular bed [17,18]. Short et al. showed that ECMO impaired cerebrovascular autoregulation in healthy newborn lambs [37]. Ingyinn et al. investigated the association between ECMO and endothelial dysfunction. They found that the middle cerebral arteries of newborn lambs exposed to ECMO had impaired myogenic response and altered endothelial function [17]. Further studies confirmed this impairment, showing an alteration in the EDNO-dependent relaxation mediated by acetylcholine stimulation, with recovery after adding a NO donor [18,38].

Possible explanations for the coronary endothelial dysfunction observed in our study, as well as for the cerebral arteries, are the hemodynamic changes induced by the veno-arterial ECMO. Indeed, the veno-arterial ECMO pump flow decreases the left ventricular output, reducing the physiological pulsatile arterial blood flow proportional to the cardiac index [39]. Moreover, a Doppler evaluation of the pericallosal artery demonstrated that the continuous non-pulsatile ECMO pump flow increases diastolic flow, exposing vessels to shear stress, which altered the endothelial production of NO, limiting the vasodilation [40,41].

In addition to hemodynamic changes, ECMO plays a direct role in developing endothelial dysfunction. The exposure to the extracorporeal circuit and related SIRS-like, transfusion burden, hyperoxia, hemolysis, and sequestration of antioxidants into the ECMO circuit is responsible for reactive oxygen species (ROS) formation [11]. Oxidative stress determines a direct endothelial injury and a phenotypic switch toward an inflammatory and apoptotic stage in the smooth muscle cells [42]. Furthermore, the proportion between ROS production and NO is the determinant of the vascular response to inflammation. In physiological conditions, NO predominance favors normal arterial vasomotor function, endothelial barrier integrity, and an anti-adhesive endothelial cell surface [43].

Conversely, the balance between NO and ROS is shifted toward the latter species during inflammation due to reduced NO biosynthesis and inactivation of available NO [44,45,46]. Oxidant excess reduces tetra-hydro-biopterin (BH4), the eNOS cofactor [47]. Insufficient concentration of BH4 inhibits the normal oxygenase function of eNOS. This triggers an active production of ROS by activating its reductase function, further favoring the harmful effects in endothelial and vascular function [48]. Additionally, ROS are responsible for the upregulation of cell adhesion molecules and chemotactic molecules through ROS-sensitive nuclear transcription factors, such as NFkB and AP-1. This is particularly important considering the exponential increase in pro-inflammatory cytokines during ECMO [49,50].

Furthermore, the endothelial dysfunction induced by oxidative stress activates the intrinsic coagulation pathway, complement system, platelet dysfunction, fibrinolysis activation, and acquired von Willebrand syndrome, with a potential procoagulant effect [11,43]. Moreover, the clot formation triggers the consumption of coagulation factors and platelets, thus, increasing the risk of disseminated intravascular coagulopathy and bleeding [10]. Further, the cannulation adds additional hemodynamic risk in terms of overload, shear stress, and hypoxia.

Other limitations of our study are worth considering. First, we only evaluated the functional impairment secondary to ECMO, but endothelial damage can be analyzed from different perspectives. In particular, the plasma analysis of vascular growth factors, stem cells, and the glycocalyx components seem to be promising markers, opening new implications for ECMO and, consequently, new therapeutic strategies [51,52]. In this regard, a recent study showed a reduction in endothelial progenitor (EPC) and mesenchymal stromal cells (MSCs), associated with a decrease in vascular endothelial growth factor (VEGF) and an increase in angiopoietin 2 (Ang-2) in ECMO-supported infants with a congenital diaphragmatic hernia [51].

Furthermore, our model started from a healthy animal, while in the clinical setting, infants requiring ECMO are always exposed to pre-ECMO mild to moderate hypoxia [39]. This probably involves a further accentuation or diversification of endothelial injury in these tiny patients. However, this model allowed us to assess the ECMO-related endothelial dysfunction specifically.

Lastly, an important limitation is related to our study’s small sample size due to the ECMO procedure’s complexity and application of the three Rs ethics principle in animal experimentation.

A better understanding of the endothelial pathways altered during ECMO could provide new insights for improving outcomes. Unfortunately, data on the long-term medical outcomes of neonatal ECMO survivors are limited and chronic cardiovascular complications, such as hypertension, are recognized mainly as secondary to chronic kidney injury [53]. For this reason, the cardiovascular outcome is not yet included in the long-term recommendations for follow-up after neonatal ECMO, focusing more on neurocognitive, motor, sensory, and growth outcomes [54]. In addition, an extensive patient cohort would be necessary to verify how treatment with extracorporeal circulation could affect chronic endothelial diseases, especially atherosclerosis. However, increasing knowledge in this area is fundamental to implementing the multidisciplinary follow-up accompanying our patients up to adolescence and beyond.

In conclusion, our findings confirm a reduction in BK-mediated vasorelaxation in the coronary arteries of piglets undergoing 8 h of ECMO, suggesting that ECMO alters EDNO-dependent coronary relaxation.

## Figures and Tables

**Figure 1 biomedicines-10-02083-f001:**
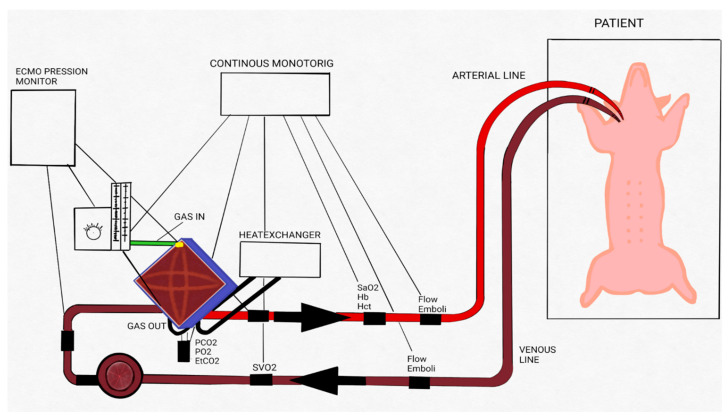
Extracorporeal membrane oxygenation (ECMO) circuit scheme. Schematic drawing of the ECMO circuit used for animal experiments. Adapted with permission from Ref. [26]. Copyright 2018, copyright owner’s Raffaeli G et al. [26]”.

**Figure 2 biomedicines-10-02083-f002:**
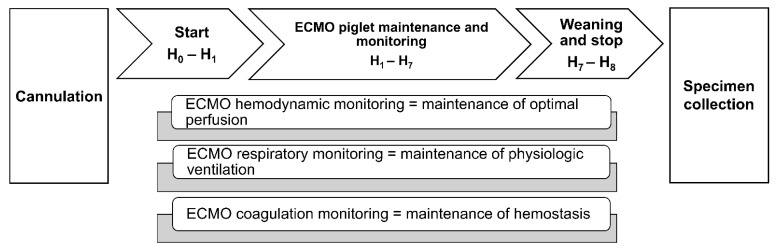
Extracorporeal membrane oxygenation (ECMO) timeline. Schematic representation of ECMO procedure’s steps and monitored parameters.

**Figure 3 biomedicines-10-02083-f003:**
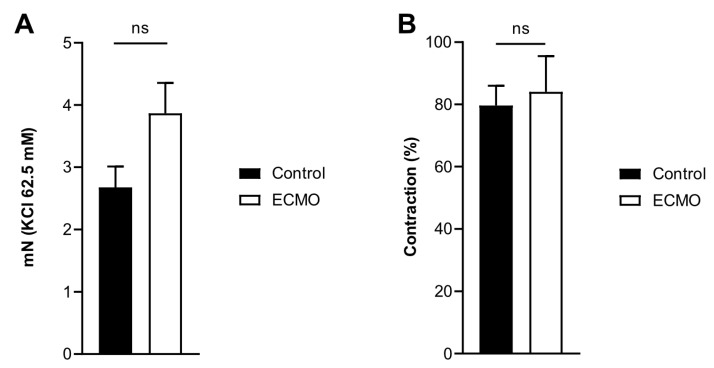
Contraction of porcine coronary arteries showed no difference between control and ECMO groups. Active wall tension induced by KCl 62.5 mM (panel **A**) and U46619 (1 μM; panel **B**) in control (*n* = 5) and ECMO (*n* = 5) groups. No significant difference was found between the two groups (panel **A**, Control: mean = 2.67, S.E.M. = 0.34; ECMO: mean = 3.87, S.E.M. = 0.49; Student’s *t*-test, *p* = 0.05; panel **B**, Control: mean = 79.64, S.E.M. = 6.36; ECMO: mean 86.07, S.E.M. = 11.48; Student’s *t*-test, *p* = 0.74). ECMO: extracorporeal membrane oxygenation. ns: not statistically significant.

**Figure 4 biomedicines-10-02083-f004:**
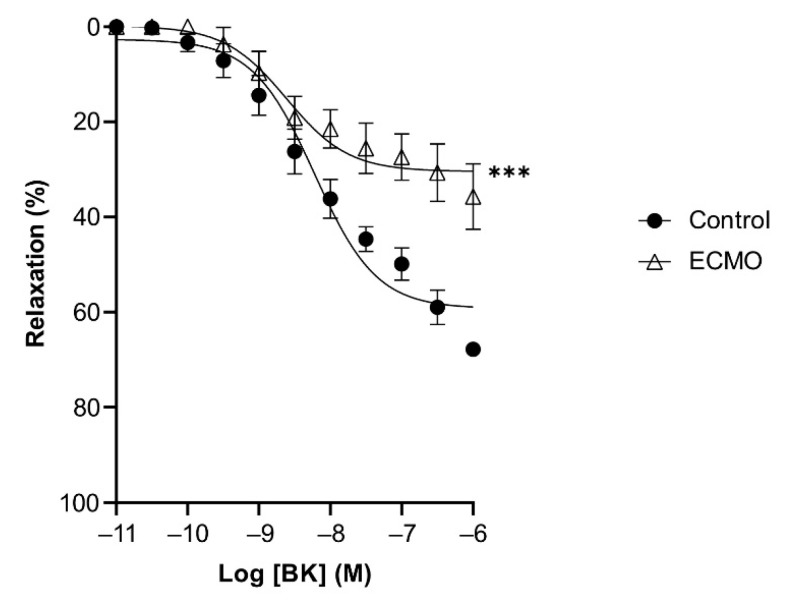
Relaxation in response to bradykinin (BK) is reduced in ECMO-treated coronary artery rings. Cumulative dose–response curves to BK in porcine coronary artery rings in control and ECMO group. Vessels were precontracted with U46619 (1 μM). Data points are presented as mean ± S.E.M (*n* = 5 in both groups). *** *p* < 0.0001 ECMO vs. Control. ECMO: extracorporeal membrane oxygenation.

**Figure 5 biomedicines-10-02083-f005:**
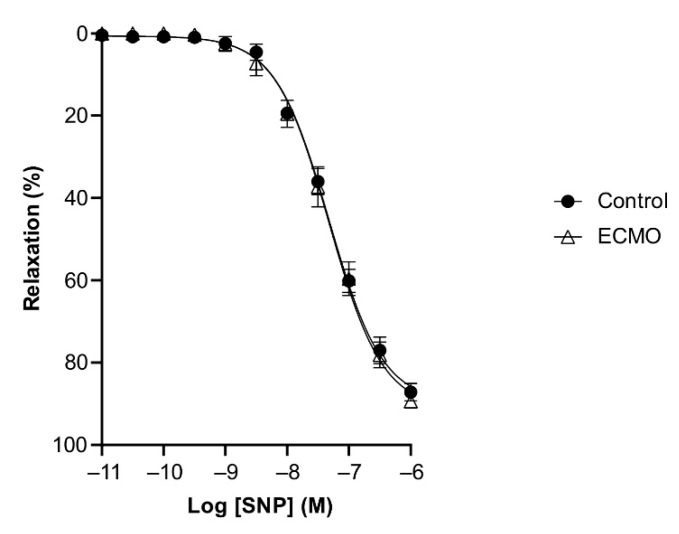
Relaxation of coronary arteries in response to sodium nitroprusside (SNP) was not affected by exposure to ECMO. Cumulative dose–response curves to SNP in porcine coronary artery rings in control and ECMO group. Vessels were precontracted with U46619 (1 μM). Data points are presented as mean ± S.E.M (*n* = 5 in both groups). ECMO: extracorporeal membrane oxygenation.

**Figure 6 biomedicines-10-02083-f006:**
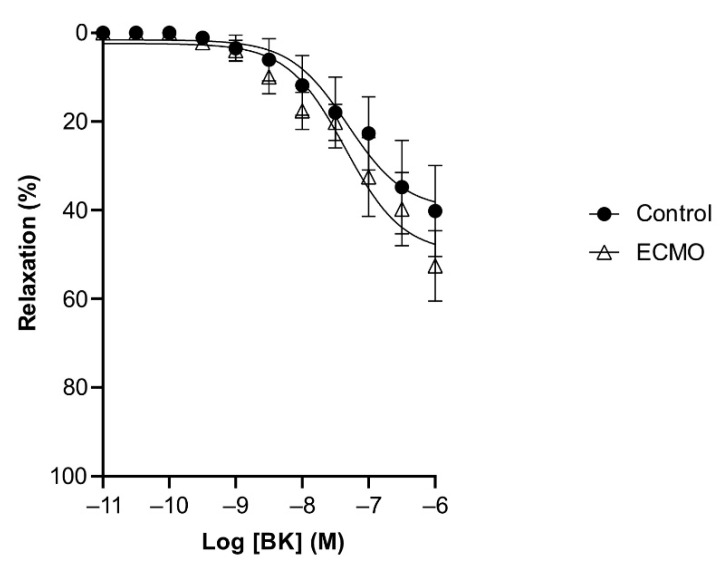
Relaxation in response to bradykinin (BK) in the presence of L-NAME showed no difference in coronary artery rings of control and ECMO groups. Cumulative dose–response curves to BK in the presence of L-NAME in porcine coronary artery rings in control and ECMO group. Vessels were precontracted with U46619 (1 μM). Data points are presented as mean ± S.E.M (*n* = 5 in both groups). ECMO: extracorporeal membrane oxygenation.

**Figure 7 biomedicines-10-02083-f007:**
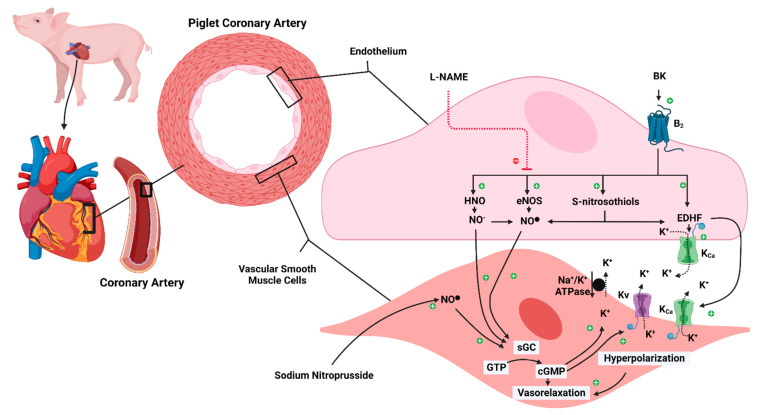
Schematic representation of ECMO-induced endothelial dysfunction. B_2_: bradykinin receptors 2; BK: bradykinin; cGMP: 3′-5′-cyclic guanosine monophosphate; ECMO: extracorporeal membrane oxygenation; ECMO: extracorporeal membrane oxygenation; EDHF: endothelium-derived hyperpolarization factor; eNOS: endothelial nitric oxide synthase; GTP: Guanosine-5′-triphosphate; HNO: nitroxyl; K^+^: potassium ion; K_Ca_: calcium-activated potassium channel; K_v_: voltage-gated potassium channel; L-NAME: Nω-nitro-L-arginine methyl ester HCl; Na^+^/K^+^ ATPase: sodium-potassium adenosine triphosphatase; NO^−^: nitroxyl anion; NO^●^: neutral free radical nitric oxide; sGC: soluble guanylate cyclase. The dashed red line has an inhibiting action. The solid black arrow has an activation action. The dashed black arrow indicates the getaway of the K ion from the cell. Created with BioRender.com.

**Table 1 biomedicines-10-02083-t001:** Physiological characteristics during extracorporeal membrane oxygenation (ECMO).

Parameters	Mean (S.E.M.)
HR (bpm)	142.3 (3.09)
SpO_2_ (%)	97.7 (0.43)
sAP (mmHg)	96.8 (2.35)
dAP (mmHg)	67.3 (2.39)
mAP (mmHg)	76.3 (2.22)
ACT (s)	211.3 (17.25)
Diuresis (mL/kg/h)	3.9 (0.57)
T (°C)	37 (0.17)
pH	7.3 (0.00)
pO_2_ (mmHg)	47.6 (1.95)
pCO_2_ (mmHg)	45.6 (1.07)
BE	−1.8 (0.46)
HCO3^−^	23.1 (0.43)
Lactate (mEq/L)	4.6 (0.26)
Hb (g/dL)	11.5 (0.26)
Hct (%)	35.4 (0.80)
K^+^ (mmol/L)	4 (0.08)
Na^+^ (mmol/L)	145.2 (0.53)
Cl^−^ (mmol/L)	107.3 (0.58)
Ca^2+^ (mmol/L)	1.4 (0.02)
Cardiac Index (mL/kg/m)	86.6 (2.32)
ven P (mmHg)	−28.4 (0.93)
int P (mmHg)	154.2 (9.97)
Art P (mmHg)	130.9 (4.24)
Sweep Gas (L/m)	0.25 (0.02)
FiO_2_ ECMO	0.4 (0.01)
SvO_2_ (%)	78.2 (0.12)
SaO_2_ (%)	98.5 (0.21)
DO_2_ (mL/m)	166.8 (4.36)
VO_2_ (mL/m)	37.2 (0.92)
O_2_ER	0.2 (0.00)
PEEP (cmH_2_O)	4.9 (0.01)
PIP (cmH_2_O)	20.5 (0.74)
RR (rpm)	31.9 (1.99)
FiO_2_ ventilator	0.2 (0.01)

ACT: activated coagulation time; Art P: arterial pressure; BE: base excess; dAP: diastolic arterial pressure; DO_2_: oxygen delivery; ECMO: extracorporeal membrane oxygenation; FiO_2_: Fraction of inspired O_2_; Hb: hemoglobin; Hct: hematocrit; HR: heart rate; int P: internal pressure; mAP: mean arterial pressure; O_2_ER: VO_2_/DO_2_ ratio; pCO_2_: partial pressure of CO_2_; PEEP: positive end-expiratory pressure; PIP: peak inspiratory pressure; pO_2_: partial pressure of O_2_; RR: respiratory rate; SaO_2_: arterial oxygen saturation; sAP: systolic arterial pressure; SpO_2_: oxygen saturation; SvO_2_: venous oxygen saturation; T: Temperature; ven P: venous pressure; VO_2_: oxygen consumption.

**Table 2 biomedicines-10-02083-t002:** Relaxant responses of porcine coronary arteries to relaxant agents.

	Control (*n* = 5)			ECMO (*n* = 5)		
	E_max_	pD_2_	*n*	E_max_	pD_2_	*n*
Bradykinin	59.21 (2.06) ***	8.22 (0.10)	5	30.41 (2.16)	8.61 (0.21)	5
Bradykinin + L-NAME	39.88 (5.50)	7.31 (0.27)	5	49.68 (4.41)	7.35 (0.17)	5
SNP	89.79 (1.77)	7.32 (0.04)	5	91.31 (2.34)	7.32 (0.05)	5

Variables are expressed as mean (S.E.M.); *n* = number of animals; E_max_= maximal relaxant effect (% of papaverine-induced relaxation); SNP: sodium nitroprusside; pD_2_ = −log EC_50_; *** *p* < 0.0001 control vs. ECMO. ECMO: extracorporeal membrane oxygenation.

## Data Availability

The data presented in this study are available on request from the corresponding author.

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
