# Peer review of "Veno-Arterial Extracorporeal Membrane Oxygenation (ECMO) Impairs Bradykinin-Induced Relaxation in Neonatal Porcine Coronary Arteries"

_biomedicines, 2022, doi:10.3390/biomedicines10092083_

Round 1

Reviewer 1 Report

This paper looks at Extracorporeal membrane oxygenation in baby piglets to determine it's effects on coronary artery function.  The paper is generally well written and the study was designed well.  However, there are  few places additional information in the methods would help the reader understand what was done and some editing needs to be performed.  Specific comments are below

page 2 line 62 suggest saying for respiratory, cardiac or cardiopulmany resuscitation respectively

page 2 lines 72-74 Might edit to say "Typically, 1-2 h after beginning the procedure, the contact between blood and circuit triggers complement....."

and what do the authors mean when they say "blood and circuit"?  They may want to specify they are talking about the blood perfusion circuit that is set up during ECMO

page 2 line 78  consider editing to "as is frequency the case in the neonatal period"

Methods

pg 3 line 132 are these numbers the external and internal diameters of the tubing?

pg 5 line 162 please add doses of each euthanasia solutions

 pg 5 under vascular reactivity studies, please add how long a tissue was perfused with the various drugs at each dose or how long a tissue was given to respond.

In the discussion, the authors list a number of limitations of their paper. Have the authors considered allowing for a recovery period after the ECMO to determine if there is a recovery in endothelial function and responsiveness to various vasomodulating substances?

Author Response

Dear Reviewer,

Thanks a lot for your commentary.

  1. Page 2, line 62: We inserted the word “respectively”, as requested
  2. Page 2, lines 72-74: The phrase was changed as requested. Moreover, we specified better the concept of the contact between blood and circuit, as requested.
  3. Page 2, line 78: we changed the phrase as requested.
  4. Page 3, line 132: The PLS Pediatric set comprises tubes whose external diameter is 1/4 inches, and the tube thickness is 1/16. Usually, it is necessary to specify the thickness of the tube, as the noninvasive sensors that are applied are calibrated according to this. However, the characteristics of the tubes are the same for all manufacturers in the neonatal setting. Nevertheless, to clarify the method better for those not used to ECMO, I have specified it better in the text as requested.
  5. Page 5, line 162: we added the euthanasia dose for medetomidine and propofol as requested.
  6. Section 2.5 Vascular reactivity studies on page 5: we specified that the contraction with KCl at 62.5 mM, U46619 at 1 µM, and papaverine hydrochloride at 0.1 mM was 10 minutes for each drug. Furthermore, in the text (lines 187-189), it is specified that each drug dose was maintained until the plateau was reached or at most 5-10 minutes if there was no effect, followed by the next incremented dose.
  7. The hypothesis is interesting, and we have discussed it among ourselves for possible other works. The current protocol, authorized by the ministerial authorities, did not provide a recovery period. Therefore, it was impossible to assess whether the endothelial dysfunction could be reduced with the suspension of ECMO support.

Reviewer 2 Report

Dear authors, thank you for the very interesting manuscript and hypothesis. As you have written, your study has a lot of limitations. However, it would be still interesting for the readers. Manuscript Veno-Arterial Extracorporeal Membrane Oxygenation (Ecmo) 2 Impairs Bradykinin-Induced Relaxation in Neonatal Porcine 3 Coronary Arteries is understandable, explanatory, and well readable. The structure is precise and clear. I have some comments and questions:

-            Row 206 – what do you mean ,,...in n animals.“ ?

-            What is the power of your statistics, when you have just 5 animals per group?

-              Why the control animals were not exposed to all procedures as animals in ECMO group? E.g. continuous anesthesia, antibiotic therapy, cystostomy.

-              In Table 1, means with SD are presented, but authors intended to use SEM (chapter „Data analysis“. Please unify and use either SD or SEM.

-            In Figure 3 you present mean and SEM but you stated that you used Mann Whitney U test – I am confused. Why have you used this test and not t-test? Are your data non-parametric? In case your data are not-parametric, please present the median and IQR.

-            Please be uniform in spacing (for example p=0.05 and p = 0.05), and in terminology (swine/ pig)

-            Figure 3 A – why so wide y-axe? A maximum of 5 would be enough

-            I am not sure if Figure 7 fits into the Results chapter - I suggest to transfer it (with lines 275-278) into Discussion section.

-            I am missing short s explanations in Figure 7

-            Please, correct the short in the title – use capitals (ECMO)

Author Response

Dear Reviewer,

Thanks a lot for your commentary.

  1. Page 5, line 207: “n animals” were deleted.
  2. We are aware that a number of 10 animals (5 in each group) would seem a small number for the achievement of an adequate sample size, but the significant results, the complexity of the ECMO procedure, and the ethical principles of the three Rs animal experimentation motivated us to publish these data anyway. Hence, we added a sentence in the discussion as a limitation study (page 12, lines 408-410).

Díaz, L., Zambrano, E., Flores, M. E., Contreras, M., Crispín, J. C., Alemán, G., ... & Bobadilla, N. A. (2021). Ethical considerations in animal research: The principle of 3R's. Revista de investigacion clinica73(4), 199-209

Bennett, R.H., Ellender, B.R., Mäkinen, T., Miya, T., Pattrick, P., Wasserman, R.J., Woodford, D.J., Weyl, O.L.F., 2016. Ethical considerations for field research on fishes. Koedoe 58.. doi:10.4102/koedoe.v58i1.1353

  1. When we decided on the study protocol, we considered the option of choosing a group of animals undergoing a surgical procedure and drugs as control vs. an ECMO group. However, this hypothesis has been discarded as literature associates the surgical procedure and the use of medications with endothelial dysfunction. Hence, control animals did not undergo any procedures to limit the influence of any drug or procedure that, by increasing stress, would subsequently alter our results. Moreover, the ECMO neonatal procedure in clinical settings is not exempt from a surgical intervention related to the introduction of cannulas and the use of sedative and inotropic drugs and transfusions. Therefore, in the clinical setting, the control cannot be other than the healthy patient not subjected to any procedure (outcomes were always compared to healthy ones). In the same way, we decided only to include healthy animals in the control group.

Riedel, B., Browne, K., & Silbert, B. (2014). Cerebral protection: inflammation, endothelial dysfunction, and postoperative cognitive dysfunction. Current Opinion in Anesthesiology, 27 (1), 89-97.

Gokce N, Keaney JF Jr, Hunter LM, et al. Risk stratification for postoperative cardiovascular events via noninvasive assessment of endothelial function: a prospective study. Circulation 2002; 105: 1567–1572.

Schier R, Hinkelbein J, Marcus H, et al. Preoperative microvascular dysfunction: a prospective, observational study expanding risk assessment strategies in major thoracic surgery. Ann Thorac Surg 2012; 94: 226–233.

Balčiūnas, M., Bagdonaitė, L., Samalavičius, R., & Baublys, A. (2009). Markers of endothelial dysfunction after cardiac surgery: soluble forms of vascular-1 and intercellular-1 adhesion molecules. Medicine, 45 (6), 434.

Taddei, S., Virdis, A., Ghiadoni, L., Sudano, I., & Salvetti, A. (2002). Effects of antihypertensive drugs on endothelial dysfunction. Drugs, 62 (2), 265-284.

  1. Table 1: data were unified as mean and S.E.M.
  2. Sorry for the mistake. The data are parametric. Then, the test used was the Student’s T-test. We corrected the legend in figure 3
  3. We have uniformed the spacing and terminology as requested.
  4. We changed the y-ax of figure 3A as recommended.
  5. We moved figure 7 into the Discussion section as recommended.
  6. We added the short explications in figure 7 as recommended.
  7. We changed in the capital the word ECMO in the title. Sorry, it was a typing error.